# Prediction of massive bleeding in pancreatic surgery based on preoperative patient characteristics using a decision tree

**Taiichi Wakiya**[1]*, **Keinosuke Ishido**[1], **Norihisa Kimura**[1], **Hayato Nagase**[1], **Shunsuke Kubota**[1], **Hiroaki Fujita**[1], **Yusuke Hagiwara**[1], **Taishu Kanda**[1], **Masashi Matsuzaka**[2], **Yoshihiro Sasaki**[2], **Kenichi Hakamada**[1]

1 Department of Gastroenterological Surgery, Hirosaki University Graduate School of Medicine, Hirosaki, Aomori, Japan, 2 Department of Medical Informatics, Hirosaki University Hospital, Hirosaki, Aomori, Japan

* wakiya1979@hirosaki-u.ac.jp

**Data Availability Statement:** All relevant data are within the manuscript and its Supporting information files.

## Abstract

Massive intraoperative blood loss (IBL) negatively influence outcomes after surgery for pancreatic ductal adenocarcinoma (PDAC). However, few data or predictive models are available for the identification of patients with a high risk for massive IBL. This study aimed to build a model for massive IBL prediction using a decision tree algorithm, which is one machine learning method. One hundred and seventy-five patients undergoing curative surgery for resectable PDAC at our facility between January 2007 and October 2020 were allocated to training (n = 128) and testing (n = 47) sets. Using the preoperatively available data of the patients (34 variables), we built a decision tree classification algorithm. Of the 175 patients, massive IBL occurred in 88 patients (50.3%). Binary logistic regression analysis indicated that alanine aminotransferase and distal pancreatectomy were significant predictors of massive IBL occurrence with an overall correct prediction rate of 70.3%. Decision tree analysis automatically selected 14 predictive variables. The best predictor was the surgical procedure. Though massive IBL was not common, the outcome of patients with distal pancreatectomy was secondarily split by glutamyl transpeptidase. Among patients who underwent PD (n = 83), diabetes mellitus (DM) was selected as the variable in the second split. Of the 21 patients with DM, massive IBL occurred in 85.7%. Decision tree sensitivity was 98.5% in the training data set and 100% in the testing data set. Our findings suggested that a decision tree can provide a new potential approach to predict massive IBL in surgery for resectable PDAC.

## Introduction

Pancreatic ductal adenocarcinoma (PDAC) has the poorest prognosis of any cancer, worldwide [1]. In tackling this lethal disease, surgery has been one of the most fundamental treatment options [2–4]. Today, pancreatic cancer surgery outcomes have improved thanks to increasing experience and refinement in surgical technique, as well as centralization of patient

**Funding:** The authors received no specific funding for this work.

**Competing interests:** The authors have declared that no competing interests exist.

care to high-volume centers [5–10]. However, due to the technical complexity of the procedure, pancreatic cancer surgery with lymph node dissection sometimes causes massive intraoperative blood loss (IBL), even when performed by experienced surgeons in high-volume centers [11]. As a result, allogeneic blood transfusion (ABT) has become commonplace for patients undergoing surgery for PDAC [11–13].

Although ABT can be a lifesaving treatment during pancreatic cancer surgery, it can cause an immunomodulatory effect called transfusion related immunomodulation [14–16]. In 1973, Opelz et al. provided initial evidence for ABT-related immunomodulation [14, 15]. Since then, though there are various potentially confounding factors to consider [17–19], many studies have reported the harmful effects of ABT on the prognosis after cancer surgery [13, 16, 20, 21]. Likewise, past reports have shown the negative effects of ABT on the long-term postoperative outcomes of PDAC patients [13, 20–24]. We also previously revealed that intraoperative ABT was strongly associated with poor prognosis in patients who underwent resection with curative intent for resectable PDAC [25]. Thus, we need to establish alternate strategies to ABT to improve the prognosis of PDAC patients further.

If we can predict massive IBL before surgery, it is possible to avoid ABT using various creative alternatives such as preoperative autologous blood storage and intraoperative acute normovolemic hemodilution. However, to date, it has not been possible to predict the occurrence of massive IBL beforehand. An accurate and robust prediction model would ultimately contribute to a better prognosis in PDAC patients. Therefore, in this study, we designed a prediction model for massive IBL in pancreatic cancer surgery. Here, we have successfully developed a user-friendly decision tree that predicts massive IBL in surgery for patients with resectable PDAC.

## Materials and methods

### Patients and study design

This single-center, retrospective, observational study was approved by the institutional ethics committee (reference no. 2020–202). This study was registered at the Japan Registry of Clinical Trials (https://jrct.niph.go.jp/, jRCT1020210001). Informed consent was obtained in the form of opt-out on our website (https://www.med.hirosaki-u.ac.jp/hospital/outline/resarch/resarch.html). This study was designed and carried out in accordance with the Declaration of Helsinki. Data has been reported in line with STROCSS 2019 criteria [26].

A total of 175 consecutive patients undergoing pancreatic surgery, with curative intent, for resectable PDAC at our facility between January 2007 and October 2020 were screened for study inclusion. Resectability status was made based on National Comprehensive Cancer Network guidelines. All patients had a confirmed pathologic diagnosis. In this study, we excluded the following cases: patients who had received neoadjuvant chemotherapy, anyone with remnant pancreatic cancer, or those with other synchronous malignancies.

### Perioperative variable selection

Patient data were extracted from the medical records at our facilities. A total of 34 perioperative variables were selected from patient records, categorized into five groups: 1) patient demographics (n = 4), 2) comorbidities (n = 8), 3) laboratory values (n = 14), 4) tumor factors (n = 5), and 5) operative factors (n = 2).

### Surgical procedures and operative management

We selected the type of pancreatic resection based on tumor location. Open pancreatoduodenectomy (PD) with lymph node dissections was usually performed on cases of pancreatic head

cancer. In cases of pancreatic body and tail cancer, open or minimally invasive distal pancreatectomy (DP) was performed with lymph node dissections. If we detected a swelling paraaortic lymph node, we generally performed paraaortic lymph node sampling during PD; whereas sampling was not routinely performed during DP. We performed a fresh frozen section analysis to confirm if the pancreatic cut-end margin was clear of residual cancer. If residual cancer was present at the pancreatic cut end margin, we cut the pancreas further to reach negative margin status. If necessary, to achieve curative resection, we performed a total pancreatectomy (TP) with lymph node dissections.

## Definition of massive IBL

In this study, we defined massive IBL as more than 20% of the estimated circulating blood volume (CBV), based on the model of Lundsgaard-Hansen [27]. We estimated the CBV using the following formula; CBV (mL) = 70 x body weight (kg). The IBL was calculated based on the in/out balance of the operative field. At our institution, any fluid loss from the abdominal cavity including ascites, bile, and lymphatics is considered to be intraoperative bleeding.

## Statistical analyses

Continuous variables were expressed as medians (ranges) and analyzed using nonparametric methods for non-normally distributed data (Mann–Whitney U-test). Categorical variables were reported as numbers (percentages) and analyzed using the chi-squared test or Fisher's exact test, as appropriate. Variables with a significant relationship to massive IBL in univariate analysis were used in a binary logistic regression model. Before inputting variables, we performed the Spearman correlation analysis and confirmed that there was no strong correlation (r >0.80) between the independent variables. Recurrence free survival (RFS) and Disease specific survival (DSS) were calculated using the Kaplan–Meier method, and differences in the survival rates between the massive IBL and non-massive IBL groups were compared using the log-rank test. RFS was defined as the time from the operation to the date of disease recurrence. DSS was defined as the time from the operation to the time of death due to PDAC, or the last follow-up time. This study was planned with a maximum follow-up period of five years. A difference was considered to be significant for values of P < 0.05. The statistical analyses were performed using IBM SPSS Statistics for Windows, Version 26.0 (IBM Corp, Armonk, NY, USA).

## Decision tree analysis

We built a tree to predict occurrence of massive IBL using classification and regression trees (CART) analysis. Enrolled patients were divided into training and testing data sets, with a ratio of about 3:1. The training data included the patients who underwent pancreatic surgery between January 2007 and June 2018. The testing data included patients who had surgery between July 2018 and October 2020. The training set was used for generating the model. Each parameter was determined by performing a grid search for those with maximum accuracy. The development environment used for decision tree analysis was Python 3.6, implemented with scikit–learn 0.20 [28]. The developed software searched the training database for the factor that most effectively predicted massive IBL occurrence and its cut-off value. In brief, the decision tree was built using the following process: 1) identification of the single variable that, when used to split the dataset into two groups ("left and right children nodes"), best-minimized impurity of massive IBL occurrence in each node, according to the Gini impurity index; 2) repetition of the splitting process within each child node leading to leaf nodes where no additional splitting achieved further reductions in node impurity. In addition, a restriction

was imposed on the tree construction such that nodes resulting from any given split needed to have at least five patients. We set a maximum tree depth of six to avoid overfitting. Nodes in binary recursive partitioning trees predict massive IBL occurrence categorically but, by evaluating node impurity, also offer associated probabilities. Finally, a massive IBL risk prediction model was created based on this analysis. Furthermore, the suitability and reproducibility of the model were validated using the testing data sets.

## Results

### Patient characteristics from the training and testing data sets

The clinical characteristics of the 175 enrolled patients are shown in Table 1.

Of the 175 patients, 128 were used for the training data, and data sets from 47 patients (26.9%) were used as the testing data. Information on IBL is presented in Table 2. Of the 175 patients, 88 patients (50.3%) were included in the massive IBL group.

### Comparison of the perioperative characteristics of the massive IBL and non-massive IBL groups

The massive IBL group demonstrated higher levels on liver function tests such as aspartate aminotransferase (AST), alanine aminotransferase (ALT), glutamyl transpeptidase (GTP), and serum total bilirubin (Table 3). This is assumed to have been caused by obstructive jaundice. There were no significant differences in the comorbidities between the groups. Although the tumor related factors were almost similar among groups, the massive IBL group had higher incidences of lymphatic metastasis (P = 0.046). Massive IBL was significantly associated with PD (78.4 vs. 42.5%, P < 0.001) and portal vein resection (25.0 vs. 6.9%, P = 0.001).

### Comparison of the postoperative outcomes of the massive IBL and non-massive IBL groups

Of the 88 patients with massive IBL, 33 (37.5%) received ABT (Table 4). The massive IBL group was associated with a higher frequency of postoperative complications (Clavien-Dindo grade ≥ 3, P = 0.001), especially in terms of the rate of pancreatic fistulas (with an International Study Group for Pancreatic Surgery (ISGPF) grade ≥ B) (20.5% vs. 6.9%, P = 0.009). Moreover, the IBL groups exhibited longer periods with regard to postoperative hospital stays (P < 0.001).

The RFS time was significantly shorter in the massive IBL group than in the non-massive IBL group (median survival time (MST), 12.4 vs. 14.5 months, P = 0.013). Likewise, the DSS was shorter in the massive IBL group (MST, 28.6 vs. 40.0 months, P = 0.1124) (Fig 1).

### Binary logistic regression analysis

To predict the occurrence of massive IBL, we performed a binary logistic regression analysis. We set the occurrence of massive IBL as the dependent variable. Significant predictor variables linked with massive IBL, which were found through a univariate analysis (P<0.05), as listed in Table 3, were entered into a binary logistic regression analysis. Before inputting predictor variables, we performed the Spearman correlation analysis and confirmed that there was no strong correlation (r >0.80) between the independent variables. As a result, we found a strong correlation between AST and ALT (r >0.90, p <0.001). Thus, we selected ALT based on the p value in a univariate analysis. There were no outliers whose predicted values exceeded ± 2SD in the measured values.

**Table 1. Patient characteristics in data sets.**

| | All cases (n = 175) | Training data (n = 128) | Testing data (n = 47) |
|---|---|---|---|
| Gender, male, n | 91 (52.0) | 65 (50.8) | 26 (55.3) |
| Age, year | 70 (43–87) | 70 (50–85) | 72 (43–87) |
| Body weight, kg | 55.9 (34.0–85.0) | 56.0 (34.0–85.0) | 55.8 (37.0–77.4) |
| Body mass index, kg/m$^2$ | 22.0 (14.1–36.3) | 22.3 (14.1–36.3) | 21.8 (17.4–32.3) |
| *Comorbidities* | | | |
| Diabetes mellitus, n | 68 (38.9) | 43 (33.6) | 25 (53.2) |
| Cancer history, n | 35 (20.0) | 21 (16.4) | 14 (29.8) |
| Hypertension, n | 70 (40.0) | 45 (35.2) | 25 (53.2) |
| Heart disease, n | 20 (11.4) | 14 (10.9) | 6 (12.8) |
| Cerebrovascular disease, n | 9 (5.1) | 5 (3.9) | 4 (8.5) |
| Hepatitis, n | 20 (11.4) | 15 (11.7) | 5 (10.6) |
| Obstructive jaundice, n | 78 (44.6) | 60 (46.9) | 18 (38.3) |
| Biliary drainage, n | 63 (36.0) | 44 (34.4) | 19 (40.4) |
| *Laboratory values* | | | |
| WBC, /μL | 5520 (2230–11020) | 5165 (2230–11020) | 5640 (3390–10430) |
| Hemoglobin, g/dL | 12.7 (7.2–16.5) | 12.7 (7.2–16.3) | 12.7 (8.8–16.5) |
| Hematocrit, % | 37.3 (22.8–46.1) | 37.2 (22.8–46.1) | 37.6 (25.3–45.9) |
| Platelets, ×10$^3$/μL | 221 (64–539) | 222 (64–513) | 214 (118–539) |
| CRP, mg/dL | 0.15 (0.01–9.59) | 0.17 (0.01–9.59) | 0.12 (0.02–5.14) |
| Albumin, g/dL | 3.9 (2.0–5.7) | 3.9 (2.0–5.7) | 3.9 (2.7–4.7) |
| Total protein, g/dL | 7.0 (4.9–8.9) | 6.8 (4.9–8.9) | 7.2 (5.9–8.0) |
| Creatinine, mg/dL | 0.67 (0.40–2.02) | 0.64 (0.40–1.43) | 0.72 (0.46–2.02) |
| AST, U/L | 28 (11–406) | 29 (11–406) | 26 (14–220) |
| ALT, U/L | 31 (9–627) | 35 (9–627) | 27 (10–175) |
| GTP, U/L | 51 (9–2579) | 65 (9–1720) | 38 (9–2579) |
| Total bilirubin, mg/dL | 0.7 (0.2–32.7) | 0.8 (0.2–32.7) | 0.6 (0.2–4.6) |
| Amylase, U/L | 74 (17–737) | 74 (17–737) | 75 (25–447) |
| CA19-9, U/mL | 87 (1–9675) | 71 (1–9675) | 152 (4–2114) |
| CEA, ng/mL | 2.7 (0.5–274) | 2.7 (0.5–37.0) | 3.2 (0.7–274) |
| *Tumor factors* | | | |
| Tumor size | | | |
| TS1 | 19 (10.9) | 15 (11.7) | 4 (8.5) |
| TS2 | 98 (56.0) | 76 (59.4) | 22 (46.8) |
| TS3 | 44 (14.1) | 29 (22.7) | 15 (31.9) |
| TS4 | 14 (8.0) | 8 (6.3) | 6 (12.8) |
| UICC 8$^{th}$ edition | | | |
| T category, n | | | |
| T1 | 18 (10.3) | 16 (12.5) | 2 (4.3) |
| T2 | 85 (48.6) | 79 (61.7) | 6 (12.8) |
| T3 | 72 (41.1) | 33 (25.8) | 39 (83.0) |
| T4 | 0 | 0 | 0 |
| N category, n | | | |
| N0 | 74 (42.3) | 49 (38.3) | 25 (53.2) |
| N1 | 64 (36.6) | 49 (38.3) | 15 (31.9) |
| N2 | 37 (21.1) | 30 (23.4) | 7 (14.9) |
| M category, n | | | |
| M0 | 162 (92.6) | 117 (91.4) | 45 (95.7) |

*(Continued)*

**Table 1.** (Continued)

| | All cases (n = 175) | Training data (n = 128) | Testing data (n = 47) |
|---|---|---|---|
| M1[†] | 13 (6.9) | 11 (8.6) | 2 (4.3) |
| UICC Stage, n | | | |
| IA | 14 (8.0) | 12 (9.4) | 2 (4.3) |
| IB | 29 (16.6) | 24 (18.8) | 5 (10.6) |
| IIA | 30 (17.1) | 12 (9.4) | 18 (38.3) |
| IIB | 60 (34.3) | 45 (35.2) | 15 (31.9) |
| III | 29 (16.6) | 24 (18.8) | 5 (10.6) |
| IV | 13 (7.4) | 11 (8.6) | 2 (4.3) |
| *Operative factors* | | | |
| Surgical procedure, n | | | |
| Pancreaticoduodenectomy | 106 (60.6) | 83 (64.8) | 23 (48.9) |
| Distal pancreatectomy | 59 (33.7) | 40 (31.3) | 19 (40.4) |
| Total pancreatectomy | 10 (5.7) | 5 (3.9) | 5 (10.6) |
| Portal vein resection, n | 28 (16.0) | 19 (14.8) | 9 (19.1) |
| Minimally invasive surgery, n | 7 (4.0) | 3 (2.3) | 4 (8.5) |

ALT, alanine aminotransferase; AST, aspartate aminotransferase; CA19-9, carbohydrate antigen 19–9; CEA, carcinoembryonic antigen; CRP, C-reactive protein; GTP, glutamyl transpeptidase; TS, tumor size; UICC, Union for International Cancer Control; WBC, white blood cell;

[†]: All of the patients were diagnosed with M1 due to positive lymph nodes other than the regional lymph nodes.

Binary logistic regression indicated that ALT and surgical procedure (DP) were significant predictors of massive IBL occurrence (Chi-Square = 48.977, df = 12, and p<0.001). All twelve predictors explained 32.5% of the variability of massive IBL occurrence. The results of Hosmer and Lemeshow was p = 0.347. ALT and surgical procedure (DP) were significant at the 5% level (ALT Wald = 4.829, p = 0.028; DP Wald = 7.454, p = 0.006). The odds ratio for ALT was 1.007 (95% confidence interval (CI): 1.001–1.014) and for DP was 0.244 (95% CI: 0.089–0.672). The model correctly predicted 65.5% of cases without massive IBL and 75.0% of cases with massive IBL, giving an overall correct prediction rate of 70.3%.

## Decision tree analysis

Decision tree analysis was carried out on the training data set using 34 variables (Fig 2). The analysis automatically selected 14 predictive variables. The best predictor in the root node was the surgical procedure. Surgical procedure (including PD or not) was selected as the variable for the initial split. Among non-PD patients, the surgical procedure DP or TP was further identified as the variable of the second split. Among the patients with TP, creatinine was

**Table 2. Information of IBL.**

| | All cases (n = 175) | Training data (n = 128) | Testing data (n = 47) |
|---|---|---|---|
| IBL, mL | 750 (50–5600) | 765 (90–3915) | 650 (50–5600) |
| IBL > 20% in CBV, n | 88 (50.3) | 68 (53.1) | 20 (42.6) |
| IBL > 1000mL, n | 60 (34.3) | 46 (35.9) | 14 (29.8) |
| ABT, n | 35 (20.0) | 24 (18.8) | 11 (23.4) |

ABT, allogeneic red blood cell transfusion; CBV, circulating blood volume; IBL, intraoperative blood loss.

**Table 3. Comparison of the perioperative characteristics of massive IBL and non-massive IBL groups.**

| | All cases (n = 175) | non-massive IBL (n = 87) | massive IBL (n = 88) | P value | Logistic regression | | |
|---|---|---|---|---|---|---|---|
| | | | | | Odds Ratio | 95% CI | P Value |
| Gender, male, n | 91 (52.0) | 40 (46.0) | 51 (58.0) | 0.113 | | | |
| Age, year | 70 (43–87) | 71 (52–85) | 69 (43–87) | 0.018 | 0.957 | 0.910–1.006 | 0.083 |
| Body weight, kg | 55.9 (34.0–85.0) | 55.9 (34.0–82.5) | 55.9 (34.7–85.0) | 0.512 | | | |
| Body mass index, kg/m² | 22.0 (14.1–36.3) | 22.2 (17.1–33.3) | 22.0 (14.1–36.3) | 0.952 | | | |
| *Comorbidities* | | | | | | | |
| Diabetes mellitus, n | 68 (38.9) | 30 (34.5) | 38 (43.2) | 0.238 | | | |
| Cancer history, n | 35 (20.0) | 21 (24.1) | 14 (15.9) | 0.174 | | | |
| Hypertension, n | 70 (40.0) | 34 (39.1) | 36 (40.9) | 0.805 | | | |
| Heart disease, n | 20 (11.4) | 10 (11.5) | 10 (11.4) | 0.978 | | | |
| Cerebrovascular disease, n | 9 (5.1) | 3 (3.4) | 6 (6.8) | 0.254 | | | |
| Hepatitis, n | 20 (11.4) | 9 (10.3) | 11 (12.5) | 0.654 | | | |
| Obstructive jaundice, n | 78 (44.6) | 26 (29.9) | 52 (59.1) | < 0.001 | 0.603 | 0.090–4.041 | 0.602 |
| Biliary drainage, n | 63 (36.0) | 21 (24.1) | 42 (47.7) | 0.001 | 1.267 | 0.229–7.009 | 0.786 |
| *Laboratory values* | | | | | | | |
| WBC, /μL | 5520 (2230–11020) | 5160 (2230–9980) | 5335 (2410–11020) | 0.227 | | | |
| Hemoglobin, g/dL | 12.7 (7.2–16.5) | 12.7 (7.2–15.9) | 12.7 (8.8–16.5) | 0.829 | | | |
| Hematocrit, % | 37.3 (22.8–46.1) | 37.4 (22.8–46.1) | 37.1 (26.7–45.8) | 0.793 | | | |
| Platelets, ×10³/μL | 221 (64–539) | 214 (96–539) | 223 (64–513) | 0.430 | | | |
| CRP, mg/dL | 0.15 (0.01–9.59) | 0.11 (0.02–9.59) | 0.23 (0.01–6.50) | 0.057 | | | |
| Albumin, g/dL | 3.9 (2.0–5.7) | 4.0 (2.5–5.7) | 3.9 (2.0–4.9) | 0.005 | 0.457 | 0.198–1.054 | 0.066 |
| Total protein, g/dL | 7.0 (4.9–8.9) | 7.1 (5.4–8.9) | 6.9 (4.9–8.1) | 0.324 | | | |
| Creatinine, mg/dL | 0.67 (0.40–2.02) | 0.67 (0.43–2.02) | 0.67 (0.40–1.69) | 0.970 | | | |
| AST, U/L | 28 (11–406) | 24 (14–287) | 34 (11–406) | 0.001 | ‡ | | |
| ALT, U/L | 31 (9–627) | 23 (9–361) | 45 (9–627) | < 0.001 | 1.007 | 1.001–1.014 | 0.028 |
| GTP, U/L | 51 (9–2579) | 30 (9–1422) | 113 (11–2579) | < 0.001 | 0.999 | 0.998–1.001 | 0.219 |
| Total bilirubin, mg/dL | 0.7 (0.2–32.7) | 0.6 (0.2–32.7) | 0.9 (0.3–24.1) | 0.015 | 0.971 | 0.869–1.085 | 0.971 |
| Amylase, U/L | 74 (17–737) | 71 (17–446) | 81 (25–737) | 0.197 | | | |
| CA19-9, U/mL | 87 (1–9675) | 62 (1–3199) | 113 (1–9675) | 0.248 | | | |
| CEA, ng/mL | 2.7 (0.5–274) | 2.7 (0.6–274) | 3.0 (0.5–23.9) | 0.272 | | | |
| *Tumor factors* | | | | | | | |
| Tumor size | | | | 0.799 | | | |
| TS1 | 19 (10.9) | 11 (12.6) | 8 (9.1) | | | | |
| TS2 | 98 (56.0) | 47 (54.0) | 51 (58.0) | | | | |
| TS3 | 44 (14.1) | 23 (26.4) | 21 (23.9) | | | | |
| TS4 | 14 (8.0) | 6 (6.9) | 8 (9.1) | | | | |
| UICC 8th edition | | | | | | | |
| T category, n | | | | 0.602 | | | |
| T1 | 18 (10.3) | 10 (11.5) | 8 (9.1) | | | | |
| T2 | 85 (48.6) | 39 (44.8) | 46 (52.3) | | | | |
| T3 | 72 (41.1) | 38 (43.7) | 34 (38.6) | | | | |
| T4 | 0 | 0 | 0 | | | | |
| N category, n | | | | 0.046 | | | |
| N0 | 74 (42.3) | 44 (50.6) | 30 (34.1) | | | | |
| N1 | 64 (36.6) | 30 (34.5) | 34 (38.6) | | 1.065 | 0.467–2.426 | 0.881 |
| N2 | 37 (21.1) | 13 (14.9) | 24 (27.3) | | 1.508 | 0.570–3.986 | 0.408 |
| M category, n | | | | 0.399 | | | |

*(Continued)*

**Table 3.** (Continued)

| | All cases (n = 175) | non-massive IBL (n = 87) | massive IBL (n = 88) | P value | Logistic regression | | |
| --- | --- | --- | --- | --- | --- | --- | --- |
| | | | | | Odds Ratio | 95% CI | P Value |
| M0 | 162 (92.6) | 82 (94.3) | 80 (90.9) | | | | |
| M1[†] | 13 (6.9) | 5 (5.7) | 8 (9.1) | | | | |
| UICC Stage, n | | | | 0.212 | | | |
| IA | 14 (8.0) | 9 (10.3) | 5 (5.7) | | | | |
| IB | 29 (16.6) | 18 (20.7) | 11 (12.5) | | | | |
| IIA | 30 (17.1) | 17 (19.5) | 13 (14.8) | | | | |
| IIB | 60 (34.3) | 28 (32.2) | 32 (36.4) | | | | |
| III | 29 (16.6) | 10 (11.5) | 19 (21.6) | | | | |
| IV | 13 (7.4) | 5 (5.7) | 8 (9.1) | | | | |
| *Operative factors* | | | | | | | |
| Surgical procedure, n | | | | < 0.001 | | | |
| Pancreaticoduodenectomy | 106 (60.6) | 37 (42.5) | 69 (78.4) | | | | |
| Distal pancreatectomy | 59 (33.7) | 47 (54.0) | 12 (13.6) | | 0.244 | 0.089–0.672 | 0.006 |
| Total pancreatectomy | 10 (5.7) | 3 (3.4) | 7 (8.0) | | 1.991 | 0.378–10.482 | 0.416 |
| Portal vein resection, n | 28 (16.0) | 6 (6.9) | 22 (25.0) | 0.001 | 2.366 | 0.816–6.864 | 0.113 |

ALT, alanine aminotransferase; AST, aspartate aminotransferase; CA19-9, carbohydrate antigen 19–9; CEA, carcinoembryonic antigen; CI, confidence interval; CRP, C-reactive protein; GTP, glutamyl transpeptidase; IBL, intraoperative blood loss; TS, tumor size; UICC, Union for International Cancer Control; WBC, white blood cell;

[†]: All of the patients were diagnosed with M1 due to positive lymph nodes other than the regional lymph nodes.

[‡]: Excluded due to multicollinearity with ALT.

identified as the variable of the third split, with an optimal cut-off value of $\leq 0.705$ mg/dL. In this node, all patients under 0.705 mg/dL experienced massive IBL.

The outcome of patients with DP was split by GTP levels, with an optimal cut-off value of $\leq 98.952$ U/L. In this node, all patients over 98.952 U/L experienced massive IBL. The outcome of other patients with DP was determined by an additional predictor such as creatinine, carbohydrate antigen 19–9 (CA19-9), total bilirubin, ALT or hematocrit.

Among the patients who underwent PD, diabetes mellitus (DM) was selected as the variable of the second split. Of the 21 patients with DM, the rate of massive IBL occurrence was 85.7%. The outcome for non-DM patients undergoing PD was determined by an additional predictor such as ALT, total protein, carcinoembryonic antigen (CEA), or age.

With all the variables in the model, the decision tree achieved an accuracy of 80.5% (sensitivity of 98.5% and specificity of 60.0%) for the training data set. In the testing data set, the decision tree achieved an accuracy of 80.9%, sensitivity of 100.0%, and specificity of 66.7%.

**Table 4. Comparison of the operative outcomes of the massive IBL and non-massive IBL groups.**

| | All cases (n = 175) | non-massive IBL (n = 87) | massive IBL (n = 88) | P value |
| --- | --- | --- | --- | --- |
| ABT, n | 35 (20.0) | 2 (2.3) | 33 (37.5) | < 0.001 |
| Postoperative complications (Clavien-dindo classification grade $\geq$ 3), n | 28 (16.0) | 6 (6.9) | 22 (25.0) | 0.001 |
| Pancreatic fistula (ISGPF grade $\geq$ B), n | 24 (13.7) | 6 (6.9) | 18 (20.5) | 0.009 |
| Delayed gastric emptying (ISGPS grade $\geq$ B), n | 18 (10.3) | 7 (8.0) | 11 (12.5) | 0.332 |
| Postoperative hospital stay, days | 18 (6–73) | 16 (6–73) | 23 (9–64) | < 0.001 |

ABT, allogeneic red blood cell transfusion; IBL, intraoperative blood loss; ISGPF, the International Study Group of Pancreatic Fistula; ISGPS, the International Study Group of Pancreatic Surgery.

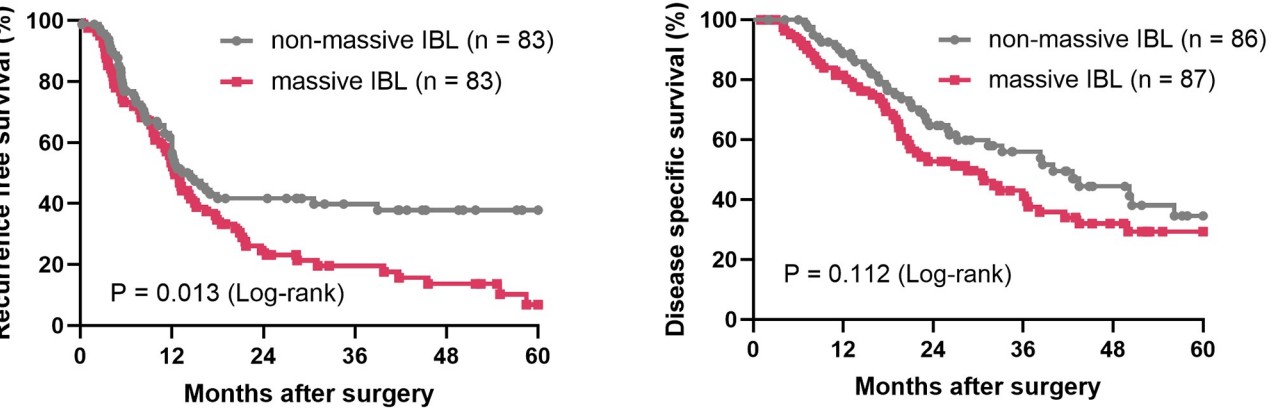

**Fig 1. Survival curves of the massive IBL and non-massive IBL groups.** IBL, intraoperative blood loss.

## Discussion

We defined the variables that could identify individuals at a risk for massive IBL in surgery for patients with resectable PDAC. Furthermore, we developed a decision tree to predict massive IBL.

The negative impact of IBL on outcomes after pancreatic surgery has long been suspected [29–34]; however, there have been few reports demonstrating risk factors for IBL [32, 33].

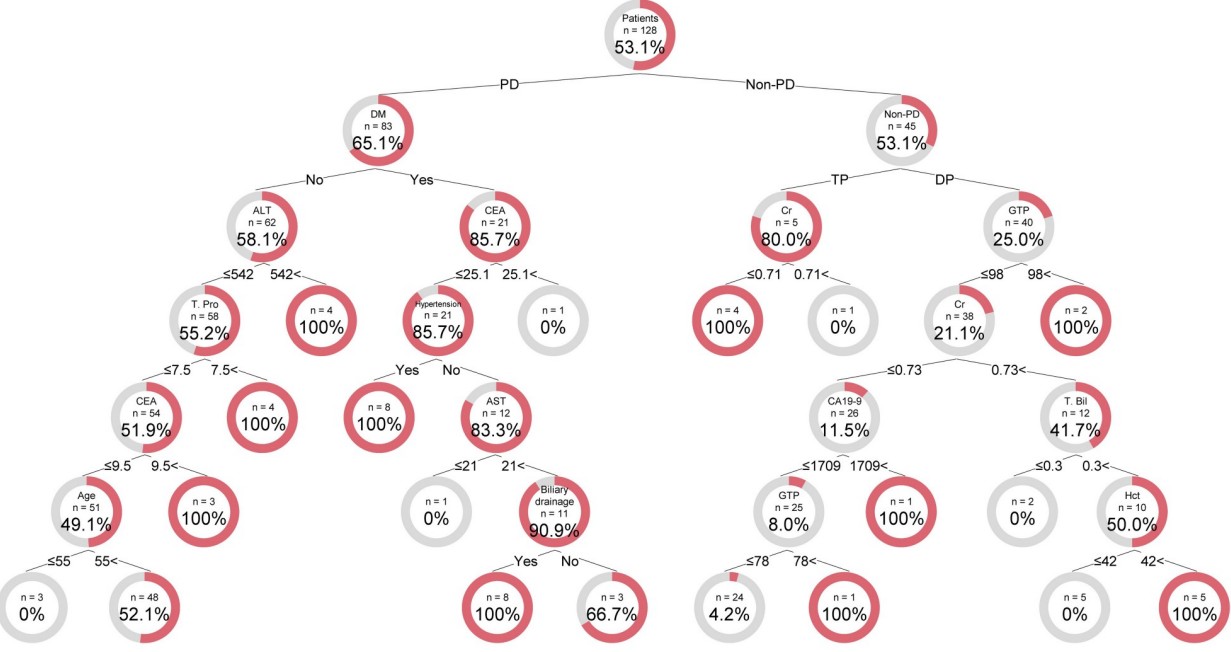

**Fig 2. Illustration of the decision tree model for massive IBL occurrence.** The sample number and factors for splitting are indicated for each node. The doughnut chart shows the percentage of patients with massive IBL (red) and without massive IBL (gray) in each node. Links between nodes indicate the cutoff value for the split or Yes/No. A high number within terminal nodes indicates that the tree would classify patients as likely to experience massive IBL. A low number in terminal nodes indicates non-massive IBL. ALT, alanine aminotransferase (U/L); AST, aspartate aminotransferase (U/L); CA19-9, carbohydrate antigen 19–9 (U/mL); CEA, carcinoembryonic antigen (ng/mL); DM, diabetes mellitus; DP, distal pancreatectomy; GTP, glutamyl transpeptidase (U/L); HCT, hematocrit (%); IBL, intraoperative blood loss; PD, pancreaticoduodenectomy; T. Bil, total bilirubin (mg/dL); TP, total pancreatectomy; T. Pro, total protein (g/dL).

Rystedt et al. retrospectively analyzed 1864 patients who had undergone a PD in the Swedish National Pancreatic and Periampullary Cancer Registry. The national study on resectable peri-ampullary tumors shows that the preoperative independent risk factors associated with major IBL ($\geq$1000 mL) were male sex, body mass index $\geq$25 kg/m$^2$, preoperative biliary drainage, C-reactive protein $\geq$12 mg/L, and neo-adjuvant chemotherapy treatment [32]. Seykora et al. conducted a multi-institutional retrospective study and precisely evaluated 5323 PD patients who had been treated for either benign or malignant disease by 62 surgeons from 17 institutions [33]. They demonstrated that factors significantly associated with increased IBL (>1300 mL) were trans-anastomotic stent placement, neoadjuvant chemotherapy, pancreaticogastrostomy reconstruction, multiorgan or vascular resection, and elevated operative time (>435 min). Furthermore, they showed that female sex, small duct ($\leq$2 mm), soft gland, minimally invasive approach, pylorus preservation, biological sealant use, and institutional volume ($\geq$67/year) were associated with decreased IBL (<300 mL). Those large cohort studies, which provided novel and significant insight to us, were analyzed using multivariable logistic regression modeling to identify the independent risk factors for massive IBL. This method has been traditionally performed in clinical studies, but there have been certain limitations, such as selecting the variables, confounding factors, and multicollinearity, as shown in this study. To resolve these problems, we attempted to make a prediction model using a decision tree analysis. This study is the first report describing a decision tree used to predict massive IBL in pancreatic surgery for resectable PDAC. The innovative value of this study is less about the excellent accuracy of the decision tree model, but more about demonstrating the potential of a novel approach for this type of prediction.

The decision tree is a machine-learning model. It comprises decision rules based on optimal feature cutoff values that recursively split independent variables into different groups and predict an outcome hierarchically [35]. The advantages of decision tree algorithms are that they are logic-based, easily interpretable, and straightforward [36]. Moreover, this machine learning method can handle both continuous and categorical variables, which suit a clinical study that includes mixed variables.

In establishing this decision tree model, surgical procedure was the first node in predicting massive IBL. Then, factors related to liver function tests, such as GTP and ALT, were usually identified as the split variable. Tumor markers, such as CA19-9 and CEA, were also identified as the split variable. These would be easily acceptable to surgeons based on their long experience.

Our model identified that hepatobiliary enzymes were risk factors for massive IBL. One of the possible explanations is that the elevation of hepatobiliary enzymes is caused by cholangitis due to bile duct obstruction. Generally, inflammation can induce neovascularization during the healing process. In the animal cholangitis model, microvessels were richly developed around the dilated bile duct [37]. It was speculated that vascular endothelial growth factor (VEGF) plays a central role in this neovascularization. Ren et al. demonstrated that overexpression of VEGF was more prominent not only in the surrounding microvessels but also in bile duct epithelium with inflammation [37]. Unfortunately, before surgery, predicting the degree of VEGF and neovascularization around the bile duct is extremely hard. Thus, it is better for us to consider patients with elevated hepatobiliary enzyme, even after biliary drainage, as at risk for massive IBL.

In this study, 88 of the 175 patients (50.3%) were included in the massive IBL group. The factors which may have led to a relatively high proportion of massive IBL are as follows. First, we defined massive IBL as more than 20% of the estimated circulating blood volume, based on the model of Lundsgaard-Hansen [27]. This definition of massive IBL is stricter than that of previous studies [32, 33]. If we define massive IBL as bleeding of over 1000 ml, 60 patients

(34.3%) would be included in the massive IBL group. Second, at our institution, any fluid loss from the abdominal cavity including ascites, bile, and lymphatics is considered to be intraoperative bleeding. Thus, only 20% of the patients required intraoperative allogeneic RBC transfusion, which is a similar rate to that of previous reports. Third, this study included only a small number of minimally invasive surgery cases. Finally, at our institution, approximately 20 different surgeons operated during the study period. Previous studies reported that surgeon volume was an important determinant of IBL [38, 39]. In short, surgeons with more experience are more likely to reduce IBL compared with their less-experienced peers. Ideally, all surgery should be performed by the most experienced surgeons. However, it is sometimes difficult to achieve this in real clinical situations. We believe that our study should be useful, especially to less-experienced surgeons and their patients.

The present study, using a decision tree, has several limitations. First, this is a retrospective single-institution cohort study. In addition, the patient population was small. If we had access to additional training data, we could achieve even higher prediction accuracy. Furthermore, we could use other machine learning methods such as some sort of neural network. Actually, we attempted the use of a neural net work and achieved an accuracy of 95.3% for the same training data set. However, the accuracy of the testing data set dropped to 54.1%. In contrast to the neural network, a decision tree visualizes a flowchart that allows appropriate treatment options for each patient depending on modifiable conditions based on that patient. Another important limitation is that the accuracy of the established model is not high enough. Thus, it would be more beneficial to focus the tree on partial data, not the entirety, and interpret them locally. To establish clinical applications, sufficient training and testing data sets are fundamental requirements for decision trees as well as neural networks. A new approach using machine learning methods that could take advantage of huge database such as national or regional data sets would be attractive for both clinicians treating PDAC and their patients.

## Conclusions

The present study, using a decision tree, has provided a new potential approach to predict massive IBL in surgery for resectable PDAC patients. To establish a more accurate prediction model for clinical application, conducting a study using a huge data set is a hope for the future.

## Supporting information

**S1 File. Our study's minimal data set.**
(TXT)

**S2 File. Source code.**
(PY)

**S3 File.**
(DOCX)

## Acknowledgments

We sincerely thank Shari Joy Berman for professionally editing the English draft of this manuscript.

## Author Contributions

**Conceptualization:** Taiichi Wakiya.

**Data curation:** Taiichi Wakiya, Keinosuke Ishido, Norihisa Kimura, Hayato Nagase, Shunsuke Kubota, Hiroaki Fujita, Yusuke Hagiwara, Taishu Kanda.

**Formal analysis:** Taiichi Wakiya, Yoshihiro Sasaki.

**Methodology:** Taiichi Wakiya, Yoshihiro Sasaki.

**Writing – original draft:** Taiichi Wakiya.

**Writing – review & editing:** Masashi Matsuzaka, Yoshihiro Sasaki, Kenichi Hakamada.

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
