## [Decision Letter · Decision Letter 0]

31 Aug 2021

PONE-D-21-25114

Prediction of massive bleeding in pancreatic surgery based on preoperative patient characteristics using a decision tree

PLOS ONE

Dear Dr. Wakiya,

Thank you for submitting your manuscript to PLOS ONE. After careful consideration, we feel that it has merit but does not fully meet PLOS ONE’s publication criteria as it currently stands. Therefore, we invite you to submit a revised version of the manuscript that addresses the points raised during the review process.

We look forward to receiving your revised manuscript.

Kind regards,

Ulrich Wellner, Prof Dr. med.

Academic Editor

PLOS ONE

Journal Requirements:

In addition, please provide further details of the generation of the decision tree and the CART algorithm in particular.

Reviewers' comments:

Reviewer's Responses to Questions

**Comments to the Author**

1. Is the manuscript technically sound, and do the data support the conclusions?

Reviewer #1: Yes

Reviewer #2: No

Reviewer #3: Yes

Reviewer #4: No

Reviewer #5: Partly

Reviewer #6: Partly

2. Has the statistical analysis been performed appropriately and rigorously? 

Reviewer #1: Yes

Reviewer #2: No

Reviewer #3: Yes

Reviewer #4: Yes

Reviewer #5: N/A

Reviewer #6: No

3. Have the authors made all data underlying the findings in their manuscript fully available?

Reviewer #1: No

Reviewer #2: No

Reviewer #3: Yes

Reviewer #4: Yes

Reviewer #5: Yes

Reviewer #6: Yes

4. Is the manuscript presented in an intelligible fashion and written in standard English?

Reviewer #1: Yes

Reviewer #2: No

Reviewer #3: Yes

Reviewer #4: Yes

Reviewer #5: No

Reviewer #6: Yes

5. Review Comments to the Author

Reviewer #1: The authors present a retrospective study examining the predictor of massive bleeding (IBL) in pancreatic surgery based on preoperative patient characteristics using a binary logistic regression analysis and a decision tree model. The best predictor was the surgical procedure using both analysis. Using the decision tree model of the training data set, the outcome of patients with DP was secondarily split by glutamyl transpeptidase. Among patients who underwent PD, DM was selected as the variable in the second split. Accuracy between the training and testing data sets was comparable (80.5% and 80.9%).

Major

1) Authors indicated that many studies have reported the harmful effects of allogeneic blood transfusion (ABT) on the prognosis after cancer surgery. Therefore, in order to prove this hypothesis, the authors should show prognosis with or without ABT.

2) In order to clarify the relevance of ABT and IBL, it is better to show the frequency of ABT by IBL.

3) Several papers describe the relationship between intraoperative bleeding and postoperative complications. The association between IBL and postoperative complications should be clarified.

4) The authors need to discuss that hepatobiliary enzymes and liver function are risk factors for IBL.

Minor

1) Figure was unclear. Would you make the figure easier to see?

Reviewer #2: 1. recent publications such as J Clin Med. 2020 Mar 4;9(3):689, J Hepatobiliary Pancreat Sci. 2016 Aug;23(8):497-507 need to be considered for citation.

2. "Of the 175 136 patients, 88 patients (50.3%) were included in the massive IBL group". It seems to be too high proportion of massive IBL. Is it so common?

3. Regarding decision tree.

1) It seems to be too complicated to apply in real clinical practice.

2) What is the rationale of decision criteria, such as DM, Cr, HT, CEA...

3) It should be presented as a form of calculator.

Reviewer #3: If intraoperative bleeding can be well predicted before surgery, the situation of these patients will be effectively improved. This study has very important clinical implications. However, if this predictive model can effectively reduce and avoid intraoperative bleeding in these patients, it would be better.

Reviewer #4: 1.Of the 175 patients, 128 were used for the training data, and data sets from 47 patients (26.9%) were used as the testing data. Based on what criteria or methodology is the grouping?

2.Massive intraoperative bleeding is a serious complication of pancreatic surgery, which is more common in the injury of the portal vein, superior mesenteric vein and superior mesenteric artery. A skilled surgeon can significantly reduce intraoperative bleeding in pancreatic surgery, so massive intraoperative bleeding is not common in large-volume centers. In your data, 88 of 175 patients experienced massive intraoperative bleeding (Of the 175 patients, 88 patients (50.3%) were included in the massive IBL group). I'm curious if all these surgeries were performed by a single surgeon? What are the causes of massive intraoperative bleeding? What were the perioperative outcomes for these patients?

Reviewer #5: Taiichi and colleagues build a model for massive IBL prediction in pancreatic surgery for PDAC by a decision tree

algorithm.The manuscript is partly technically sound and builds on current data. I have few comments here:

1.Language should be revised (seeking professional assistance is suggested).

2. The sample size described in current cohort is too small to make a strong conclusion.

3. The authors should show how the massive IBL in pancreatic surgery influence the short-term and long-term outcomes for PDAC in current data.

4.The authors defined massive IBL as more than 20% of the estimated circulating blood volume. I have two questions: First, Is there any referrence for this definition or the author made it by themselves? Second, How the authours calculate the IBL volume? Besides, of the 175 patients, 88 patients (50.3%) were included in the massive IBL group, which is a relatively high proportion.

5.The result showed that distal pancreactomy (DP) were significant predictors of massive IBL occurrence and surgical procedure was the first node in predicting massive IBL. The authours should make further discussion. Tumors located in body or tail more easily lead to left-side portal hypertension which is casued by splenic vein obstruction. Therefore, these patients are more likely to have massive IBL occurrence.

6. Alanine aminotransferase or liver function was one of significant predictors of massive IBL,however, the authors should show the coagulation function in the data and analyze its impact on IBL.

7.The discussionare too short and the limitation section are too long.

Reviewer #6: This is an impressive paper on the development of a decision tree based prediction model for intraoperative blood loss.

While the concept of the paper is interesting, I think there are a few critical problems with it. Needless to say, the performance of surgical procedures has improved over time. I think the main reasons for this are energy devices, laparoscopic surgery, and robotics. The amount of blood loss depends on what kind of energy device is used. Also, a major advantage of minimally invasive surgery is the reduction of blood loss, and I don't know if it is arbitrary that this paper does not take this into account, but it greatly reduces the value of this study. More to the point, the fact that about half of the patients in the authors' cohort had massive bleeding is problematic for the quality of surgery today. Unfortunately, I don't think that a study done with such surgical quality can provide universal facts.

6. PLOS authors have the option to publish the peer review history of their article (what does this mean?). If published, this will include your full peer review and any attached files.

Reviewer #1: No

Reviewer #2: No

Reviewer #3: No

Reviewer #4: No

Reviewer #5: No

Reviewer #6: No

---

## [Author Response · Author response to Decision Letter 0]

11 Oct 2021

Ulrich Wellner

Academic Editor

PLOS ONE

Dear Professor Wellner,

On behalf of my co-authors, I would like to express my gratitude to the Editor and the Reviewers for their kind attention to our initial manuscript, PONE-D-21-25114, entitled " Prediction of massive bleeding in pancreatic surgery based on preoperative patient characteristics using a decision tree" We greatly appreciate the Editor and Reviewers’ thoughtful and constructive comments, for they provided important scientific suggestions. These suggestions helped us enormously in preparing our new manuscript.

After careful consideration of the comments for PONE-D-21-25114, we have prepared a new manuscript in which we have improved the quality of the rendering of our data with additional results that reinforce our original findings, and we have extensively revised our manuscript. Our responses to each of the Editor and Reviewers’ comments are delineated below.

Revision notes

Responses to the Journal Requirements

We greatly appreciate your thoughtful and constructive comments. These suggestions have helped us enormously in preparing our new manuscript. We have revised our manuscript based on your comments (specific details below).

Journal Requirement 1:

Our Response to Journal Requirement 1:

We have revised the manuscript based on PLOS ONE's style requirements, including those for file naming.

Journal Requirement 2:

We suggest you thoroughly copyedit your manuscript for language usage, spelling, and grammar. If you do not know anyone who can help you do this, you may wish to consider employing a professional scientific editing service.

Our Response to Journal Requirement 2:

We have made revisions based on this Comment. This revised manuscript was edited by Shari Joy Berman, who is a professional English editor.

Journal Requirement 3:

Please note that PLOS ONE has specific guidelines on code sharing for submissions in which author-generated code underpins the findings in the manuscript. In these cases, all author-generated code must be made available without restrictions upon publication of the work.

In addition, please provide further details of the generation of the decision tree and the CART algorithm in particular.

Our Response to Journal Requirement 3:

As per this requirement, we have uploaded the code we generated as Supporting Information. The development environment used for decision tree analysis was Python 3.6, implemented with scikit–learn 0.20 (Diverted from Scikit-learn: Machine Learning in Python, Pedregosa et al., JMLR 12, pp. 2825-2830, 2011.)

Journal Requirement 4:

In your Data Availability statement, you have not specified where the minimal data set underlying the results described in your manuscript can be found. PLOS defines a study's minimal data set as the underlying data used to reach the conclusions drawn in the manuscript and any additional data required to replicate the reported study findings in their entirety. All PLOS journals require that the minimal data set be made fully available.

Our Response to Journal Requirement 4:

As suggested we have included the minimal data set in Supporting Information.

Responses to the comments from the Reviewer 1

We greatly appreciate your thoughtful and constructive comments. These suggestions helped us enormously in preparing our new manuscript. We have revised our manuscript based on your comments (explained below).

Reviewer 1, Comment 1:

Authors indicated that many studies have reported the harmful effects of allogeneic blood transfusion (ABT) on the prognosis after cancer surgery. Therefore, in order to prove this hypothesis, the authors should show prognosis with or without ABT.

Our Response to Reviewer 1, Comment 1:

Our latest article, which was accepted on September 14, 2021, reported that intraoperative ABT was strongly associated with poor prognosis in patients who underwent resection with curative intent for resectable PDAC. The RFS time was significantly shorter in the ABT group than in the non-ABT group (median survival time (MST), 10.6 vs. 14.2 months, P = 0.002). Likewise, the DSS was significantly shorter in the ABT group (MST, 20.5 vs. 38.3 months, P = 0.014). Results remained similar after propensity score matching analysis when confounding factors, other than ABT, were excluded. On page 4, lines 40-42, in the Introduction section, the following sentence was added to the text: “We also previously revealed that intraoperative ABT was strongly associated with poor prognosis in patients who underwent resection with curative intent for resectable PDAC (Kanda T, Wakiya T, Ishido K, Kimura N, Nagase H, Kubota S, et al. Intraoperative Allogeneic Red Blood Cell Transfusion Negatively Influences Prognosis After Radical Surgery for Pancreatic Cancer: A Propensity Score Matching Analysis. Pancreas. 2021. in press.).” We greatly appreciate the constructive comment and have added this to strengthen our introduction.

Reviewer 1, Comment 2:

In order to clarify the relevance of ABT and IBL, it is better to show the frequency of ABT by IBL.

Our Response to Reviewer 1, Comment 2:

Of the 88 patients with massive IBL, 33 (37.5%) received ABT. We have created a new table giving the information on ABT. On page 15, in the Results section, we have added Table 4. We greatly appreciate this constructive comment.

Reviewer 1, Comment 3:

Several papers describe the relationship between intraoperative bleeding and postoperative complications. The association between IBL and postoperative complications should be clarified.

Our Response to Reviewer 1, Comment 3:

On page 16, lines 176-181, in the Results, the following sentence was added to the text: “The massive IBL group was associated with a higher frequency of postoperative complications (Clavien-Dindo grade ≥ 3, P = 0.001), especially in terms of the rate of pancreatic fistulas (with an International Study Group for Pancreatic Surgery (ISGPF) grade ≥ B) (20.5% vs. 6.9%, P = 0.009). Moreover, the IBL groups exhibited longer periods with regard to postoperative hospital stays (P < 0.001).” We greatly appreciate this constructive comment.

Reviewer 1, Comment 4:

The authors need to discuss that hepatobiliary enzymes and liver function are risk factors for IBL.

Our Response to Reviewer 1, Comment 4:

On page 21, lines 285-295, in the Discussion section, the following sentence was added to the text: “Our model identified that hepatobiliary enzymes were risk factors for massive IBL. One of the possible explanations is that the elevation of hepatobiliary enzymes is caused by cholangitis due to bile duct obstruction. Generally, inflammation can induce neovascularization during the healing process. In the animal cholangitis model, microvessels were richly developed around the dilated bile duct [37]. It was speculated that vascular endothelial growth factor (VEGF) plays a central role in this neovascularization. Ren et al. demonstrated that overexpression of VEGF was more prominent not only in the surrounding microvessels but also in bile duct epithelium with inflammation [37]. Unfortunately, before surgery, predicting the degree of VEGF and neovascularization around the bile duct is extremely hard. Thus, it is better for us to consider patients with elevated hepatobiliary enzyme, even after biliary drainage, as at risk for massive IBL.” 

Reviewer 1, Comment 5:

Figure was unclear. Would you make the figure easier to see?

Our Response to Reviewer 1, Comment 5:

We have made the figure easier to read. 

Responses to the comments from the Reviewer 2

We greatly appreciate your thoughtful and constructive comments. These suggestions helped us enormously in preparing our new manuscript. We have revised our manuscript based on your comments (explained below).

Reviewer 2, Comment 1:

recent publications such as J Clin Med. 2020 Mar 4;9(3):689, J Hepatobiliary Pancreat Sci. 2016 Aug;23(8):497-507 need to be considered for citation.

Our Response to Reviewer 2, Comment 1:

We have added the suggested citation based on this comment.

Reviewer 2, Comment 2:

"Of the 175 136 patients, 88 patients (50.3%) were included in the massive IBL group". It seems to be too high proportion of massive IBL. Is it so common?

Our Response to Reviewer 2, Comment 2:

In this study, we defined massive IBL as more than 20% of the estimated circulating blood volume, based on the model of Lundsgaard-Hansen (Lundsgaard-Hansen P. Component therapy of surgical hemorrhage: red cell concentrates, colloids and crystalloids. Bibliotheca haematologica. 1980(46):147-69.) This definition of massive IBL may be stricter than previous studies. If we define massive IBL as bleeding of over 1000 ml, 60 patients (34.3%) would be included the massive IBL group. 

Furthermore, at our institution, any fluid loss from the abdominal cavity including ascites, bile, and lymphatics is considered intraoperative bleeding. This policy may have led to a relatively high proportion of massive IBL. Thus, only 20% of the patients required intraoperative allogeneic RBC transfusion, which is a similar rate to those in previous reports.

At our institution, though there was a difference in the number, approximately 20 different surgeons operated during the study period. These factors also may have led to a relatively high proportion of massive IBL. Previous studies reported that surgeon volume was an important determinant of IBL. In short, surgeons with more experience are more likely to reduce IBL compared with their less-experienced peers. (Ref.1: Casciani F, Trudeau MT, Asbun HJ, Ball CG, Bassi C, Behrman SW, et al. Surgeon experience contributes to improved outcomes in pancreatoduodenectomies at high risk for fistula development. Surgery. 2021;169(4):708-20. Ref-2: Schmidt CM, Turrini O, Parikh P, House MG, Zyromski NJ, Nakeeb A, et al. Effect of Hospital Volume, Surgeon Experience, and Surgeon Volume on Patient Outcomes After Pancreaticoduodenectomy: A Single-Institution Experience. Archives of Surgery. 2010;145(7):634-40.) Taken together, our study must be useful especially for less-experienced surgeons and their patients.

We have added an explanation of the high proportion of massive IBL and the implications of this study to the Discussion section.

Reviewer 2, Comment 3:

Regarding decision tree.

1) It seems to be too complicated to apply in real clinical practice.

2) What is the rationale of decision criteria, such as DM, Cr, HT, CEA...

3) It should be presented as a form of calculator.

Our Response to Reviewer 2, Comment 3:

1) We have made the figures easier to read.

2) As we described in the Materials and Methods section, each parameter was determined by performing a grid search in Scikit-learn for those with maximum accuracy. A grid search is a tuning technique that attempts to compute the optimum values of hyperparameters. It is an exhaustive search that is performed on the specific parameter values of a model. As a result, criteria such as DM, Cr, HT, CEA...were selected without being arbitrary.

3) We totally agree with you. A calculator format would be more convenient. One of the advantages of our decision tree model is that the outputs are easy to read and interpret without requiring statistical knowledge or special equipment such as computers. Ideally, in the future, we would also like to provide this in the form of app. However, at the moment, although we consider the possibility, it was too complex for us to construct the development environment of an app. Instead, we decided to share the code so that anyone could use it.

Responses to the comments from the Reviewer 3

Reviewer 3, Comment 1:

If intraoperative bleeding can be well predicted before surgery, the situation of these patients will be effectively improved. This study has very important clinical implications. However, if this predictive model can effectively reduce and avoid intraoperative bleeding in these patients, it would be better.

Our Response to Reviewer 3, Comment 1:

We totally agree with you. Our prediction model can contribute to reduce and avoid allogenic blood transfusion with various creative alternatives such as preoperative autologous blood storage. However, as you said, our ideal goal is to reduce and avoid intraoperative bleeding. Thus, based upon our decision tree, we need to conduct prospective trials attempting to intervene using modifiable factors, such as DM and cholangitis, in the future. We greatly appreciate your thoughtful and constructive comments. These comments will help us enormously in conducting our future projects.

Responses to the comments from the Reviewer 4

We greatly appreciate your thoughtful and constructive comments. These suggestions helped us enormously in preparing the new manuscript. We have revised our manuscript based on your comments (explained below).

Reviewer 4, Comment 1:

Of the 175 patients, 128 were used for the training data, and data sets from 47 patients (26.9%) were used as the testing data. Based on what criteria or methodology is the grouping?

Our Response to Reviewer 4, Comment 1:

On page 8, lines 114-116, in the Materials and Methods section, the following sentences were added to the text: “The training data included the patients who underwent pancreatic surgery between January 2007 and June 2018. The testing data included patients who had surgery between July 2018 and October 2020.” We aimed to divide the patients into training and testing data sets, with a ratio of about 3:1.

Reviewer 4, Comment 2:

Massive intraoperative bleeding is a serious complication of pancreatic surgery, which is more common in the injury of the portal vein, superior mesenteric vein and superior mesenteric artery. A skilled surgeon can significantly reduce intraoperative bleeding in pancreatic surgery, so massive intraoperative bleeding is not common in large-volume centers. In your data, 88 of 175 patients experienced massive intraoperative bleeding (Of the 175 patients, 88 patients (50.3%) were included in the massive IBL group). I'm curious if all these surgeries were performed by a single surgeon? What are the causes of massive intraoperative bleeding? What were the perioperative outcomes for these patients?

Our Response to Reviewer 4, Comment 2:

We greatly appreciate this surgically professional comment. As you have said, massive intraoperative bleeding is more common in injuries to the major vessels such as PV, SMV, and SMA. In fact, the massive IBL group had portal vein resection performed more frequently. Furthermore, half of the massive IBL group received biliary drainage preoperatively, which is considered to be an independent predictor for major intraoperative bleeding (Reference: HPB (Oxford). 2019 Mar;21(3):268-274.). At our institution, though there was a difference in the number, approximately 20 different surgeons operated during the study period. These factors may have affected the massive IBL group. 

The massive IBL group was associated with a higher frequency of postoperative complications (Clavien-Dindo grade ≥ 3, P = 0.001), especially in terms of the rate of pancreatic fistulas (with an International Study Group for Pancreatic Surgery (ISGPF) grade ≥ B) (20.5% vs. 6.9%, P = 0.009). Moreover, the IBL groups exhibited longer periods with regard to postoperative hospital stays (P < 0.001). The RFS time was significantly shorter in the massive IBL group than in the non-massive IBL group (median survival time (MST), 12.4 vs. 14.5 months, P = 0.013). Likewise, the DSS was shorter in the massive IBL group (MST, 28.6 vs. 40.0 months, P = 0.1124)

Based upon your comment, we added this information to the revised manuscript. We greatly appreciate this constructive feedback.

Responses to the comments from the Reviewer 5

We greatly appreciate your thoughtful and constructive comments. These suggestions helped us enormously in preparing our new manuscript. We have revised our manuscript based on your comments (explained below).

Reviewer 5, Comment 1:

Language should be revised (seeking professional assistance is suggested).

Our Response to Reviewer 5, Comment 1:

This revised manuscript was edited by Shari Joy Berman, who is a professional English editor.

Reviewer 5, Comment 2:

The sample size described in current cohort is too small to make a strong conclusion.

Our Response to Reviewer 5, Comment 2:

We totally agree with you. The patient population was too small to come to a clear or definite conclusion. On the other hand, our study demonstrated better prediction accuracy than traditional methods such as binary logistic regression analysis. Though the sample size was small, our study actually showed the usefulness of a decision tree and the machine learning approach. This approach has extendibility and scalability in this clinical situation of reducing massive IBL in pancreatic surgery. Your comment has helped us in revising the manuscript to state the limitations of the small cohort clearly.

Reviewer 5, Comment 3:

The authors should show how the massive IBL in pancreatic surgery influence the short-term and long-term outcomes for PDAC in current data.

Our Response to Reviewer 5, Comment 3:

Regarding short-term outcome, the massive IBL group was associated with a higher frequency of postoperative complications (Clavien-Dindo grade ≥ 3, P = 0.001), especially in terms of the rate of pancreatic fistulas (with an International Study Group for Pancreatic Surgery (ISGPF) grade ≥ B) (20.5% vs. 6.9%, P = 0.009). Moreover, the IBL groups exhibited longer periods with regard to postoperative hospital stays (P < 0.001).

Regarding long-term outcome, the RFS time was significantly shorter in the massive IBL group than in the non-massive IBL group (median survival time (MST), 12.4 vs. 14.5 months, P = 0.013). Likewise, the DSS was shorter in the massive IBL group (MST, 28.6 vs. 40.0 months, P = 0.1124). Based upon your comment, we have added this information to the revised manuscript. We greatly appreciate this constructive comment.

Reviewer 5, Comment 4:

The authors defined massive IBL as more than 20% of the estimated circulating blood volume. I have two questions: First, Is there any referrence for this definition or the author made it by themselves? Second, How the authours calculate the IBL volume? Besides, of the 175 patients, 88 patients (50.3%) were included in the massive IBL group, which is a relatively high proportion.

Our Response to Reviewer 5, Comment 4:

First, we defined massive IBL as more than 20% of the estimated circulating blood volume, based on the model established by Lundsgaard-Hansen (Lundsgaard-Hansen P. Component therapy of surgical hemorrhage: red cell concentrates, colloids and crystalloids. Bibliotheca haematologica. 1980(46):147-69.) This definition of massive IBL may be stricter than other previous studies. If we define massive IBL as bleeding of over 1000 ml, 60 patients (34.3%) would be included in the massive IBL group. 

Second, the IBL was calculated based on the “in/out” balance of the operative field. At our institution, any fluid loss from the abdominal cavity including ascites, bile, and lymphatics is considered to be intraoperative bleeding. This policy may have led to a relatively high proportion of massive IBL. Thus, only 20% of the patients required intraoperative allogeneic RBC transfusion, which is a similar rate to those in previous reports.

Furthermore, at our institution, though there was a difference in the number, approximately 20 different surgeons operated during the study period. These factors also may have led to a relatively high proportion of massive IBL. Previous studies reported that surgeon volume was an important determinant of IBL. In short, more experienced surgeons are more likely to achieve reduced IBL rates compared with their peers. (Ref.1: Casciani F, Trudeau MT, Asbun HJ, Ball CG, Bassi C, Behrman SW, et al. Surgeon experience contributes to improved outcomes in pancreatoduodenectomies at high risk for fistula development. Surgery. 2021;169(4):708-20. Ref-2: Schmidt CM, Turrini O, Parikh P, House MG, Zyromski NJ, Nakeeb A, et al. Effect of Hospital Volume, Surgeon Experience, and Surgeon Volume on Patient Outcomes After Pancreaticoduodenectomy: A Single-Institution Experience. Archives of Surgery. 2010;145(7):634-40.) Ideally, all surgery should be performed by the most experienced surgeons. However, it is sometimes difficult to achieve this in real clinical situations. We believe that our study should be useful, especially for less-experienced surgeons and their patients.

We have added an explanation of the high proportion of massive IBL and the implications of this study to the Discussion section.

Reviewer 5, Comment 5:

The result showed that distal pancreactomy (DP) were significant predictors of massive IBL occurrence and surgical procedure was the first node in predicting massive IBL. The authours should make further discussion. Tumors located in body or tail more easily lead to left-side portal hypertension which is casued by splenic vein obstruction. Therefore, these patients are more likely to have massive IBL occurrence.

Our Response to Reviewer 5, Comment 5:

We totally agree with your comments from the surgical perspective. As you have said, we sometimes operated on PDAC patients with left-side portal hypertension due to splenic vein obstruction. Our results demonstrated that when the surgical procedure (PD or not) was selected that created the initial split. Moreover, among non-PD cases, the surgical procedure (DP or TP) was identified as the second split. As a matter of fact, if anything, distal pancreatectomy was identified as not being among the significant predictors of massive IBL. We deeply apologize for confusing you about the first node. We have made the figures easier to read. 

Reviewer 5, Comment 6:

Alanine aminotransferase or liver function was one of significant predictors of massive IBL,however, the authors should show the coagulation function in the data and analyze its impact on IBL.

Our Response to Reviewer 5, Comment 6:

We totally agree with you. Just as you mention, we have really wanted to analyze the impact of coagulation function on IBL. However, particularly with the earlier cases of the study period, we did not measure the coagulation function test routinely in PDAC patients. Thus, in this study, it was difficult to use the coagulation function for machine learning due to too many missing values.

Our model identified that hepatobiliary enzymes were one significant predictor of massive IBL. One of the possible explanations is that the elevation of hepatobiliary enzymes was caused by cholangitis due to bile duct obstruction. Generally, inflammation can induce neovascularization during the healing process. In the animal cholangitis model, microvessels were richly developed around the dilated bile duct (33). It was speculated that vascular endothelial growth factor (VEGF) plays a central role in this neovascularization. This report demonstrated that overexpression of VEGF was more prominent not only in the surrounding microvessels but also in bile duct epithelium with inflammation (33). Unfortunately, before surgery, predicting the degree of VEGF and neovascularization around the bile duct is extremely hard. Thus, it is better for us to consider patients with elevated hepatobiliary enzymes, even after biliary drainage, as at risk for massive IBL. We added this information to the revised manuscript. We greatly appreciate your thoughtful and constructive comments.

Reviewer 5, Comment 7:

The discussion are too short and the limitation section are too long.

Our Response to Reviewer 5, Comment 7:

We agree with your assessment. We have revised the Discussion section.

Responses to the comments from the Reviewer 6

We greatly appreciate your thoughtful and constructive comments. These suggestions helped us enormously in preparing our new manuscript. We have revised our manuscript based on your comments (explained below).

Reviewer 6, Comment 1:

This is an impressive paper on the development of a decision tree based prediction model for intraoperative blood loss.

While the concept of the paper is interesting, I think there are a few critical problems with it. Needless to say, the performance of surgical procedures has improved over time. I think the main reasons for this are energy devices, laparoscopic surgery, and robotics. The amount of blood loss depends on what kind of energy device is used. Also, a major advantage of minimally invasive surgery is the reduction of blood loss, and I don't know if it is arbitrary that this paper does not take this into account, but it greatly reduces the value of this study. More to the point, the fact that about half of the patients in the authors' cohort had massive bleeding is problematic for the quality of surgery today. Unfortunately, I don't think that a study done with such surgical quality can provide universal facts.

Our Response to Reviewer 6, Comment 1:

We greatly appreciate such professional comments from surgical perspective. As you have said, intraoperative blood loss has decreased by the progress of energy devices and minimally invasive surgery. We deeply apologize for confusing you about this point due to the lack of information we provided initially.

In this study period, we mainly used scissors and monopolar electrosurgery. At the last part of the study period, we occasionally used ultrasonic systems (Harmonic® scalpel) or electrothermal bipolar-activated devices (LigaSure™). Furthermore, we performed minimally invasive surgery only in 4% (7cases) of this study cohort. At the beginning of the use of minimally invasive surgery methods in our institution, we mostly performed minimally invasive surgery on patients with benign disease. After the learning curve of the approach expanded, we extended the indication to PDAC. That is why this study, based on when the surgeries were originally performed, included only a small number of minimally invasive cases.

We defined massive IBL as more than 20% of the estimated circulating blood volume, based on the model of Lundsgaard-Hansen (Lundsgaard-Hansen P. Component therapy of surgical hemorrhage: red cell concentrates, colloids and crystalloids. Bibliotheca haematologica. 1980(46):147-69.) This definition of massive IBL may be stricter than previous studies. If we were to define massive IBL as bleeding of over 1000 ml, 60 patients (34.3%) would be included in the massive IBL group. 

Additionally, at our institution, any fluid loss from the abdominal cavity including ascites, bile, and lymphatics is considered to be intraoperative bleeding. This policy may have led to a relatively high proportion of massive IBL. Thus, only 20% of the patients required intraoperative allogeneic RBC transfusion, which is a similar rate to that of previous reports.

At our institution, though there was a difference in the number, approximately 20 different surgeons operated during the study period. This factor also may have led to a relatively high proportion of massive IBL. Previous studies reported that surgeon volume was an important determinant of IBL. In short, surgeons with more experience are more likely to reduce IBL compared with their less-experienced peers. (Ref.1: Casciani F, Trudeau MT, Asbun HJ, Ball CG, Bassi C, Behrman SW, et al. Surgeon experience contributes to improved outcomes in pancreatoduodenectomies at high risk for fistula development. Surgery. 2021;169(4):708-20. Ref-2: Schmidt CM, Turrini O, Parikh P, House MG, Zyromski NJ, Nakeeb A, et al. Effect of Hospital Volume, Surgeon Experience, and Surgeon Volume on Patient Outcomes After Pancreaticoduodenectomy: A Single-Institution Experience. Archives of Surgery. 2010;145(7):634-40.) Ideally, all surgery should be performed by the most experienced surgeons. However, it is sometimes difficult to achieve this in real clinical situations. We believe that our study should be useful, especially for less-experienced surgeons and their patients.

We have added an explanation of the high proportion of massive IBL and the implications of this study to the Discussion section.

We hope that you find our revised manuscript suitable for publication in PLOS ONE.

In closing, we would like to thank you once again for your reviews and suggestions, which have helped us improve the quality of our paper. We would greatly appreciate your kind consideration for our revised manuscript.

Sincerely yours,

Taiichi Wakiya, M.D., Ph.D.

Department of Gastroenterological Surgery

Hirosaki University Graduate School of Medicine

5, Zaifu-cho, Hirosaki, Aomori, 036-8562, Japan

Telephone: +81-172-395079, Fax: +81-172-395080

E-mail: wakiya1979@hirosaki-u.ac.jp

---

## [Editor Report · Decision Letter 1]

25 Oct 2021

Prediction of massive bleeding in pancreatic surgery based on preoperative patient characteristics using a decision tree

PONE-D-21-25114R1

Dear Dr. Wakiya,

We’re pleased to inform you that your manuscript has been judged scientifically suitable for publication and will be formally accepted for publication once it meets all outstanding technical requirements.

Kind regards,

Ulrich Wellner, Prof Dr. med.

Academic Editor

PLOS ONE
---

## [Editor Report · Acceptance letter]

29 Oct 2021

PONE-D-21-25114R1 

Prediction of massive bleeding in pancreatic surgery based on preoperative patient characteristics using a decision tree 

Dear Dr. Wakiya:

I'm pleased to inform you that your manuscript has been deemed suitable for publication in PLOS ONE. Congratulations! Your manuscript is now with our production department. 

Kind regards, 

on behalf of

Mr. Ulrich Wellner 

Academic Editor

PLOS ONE